Review

Subject Area:
biochemistry/microbiology

Keywords:
translation, antibiotic resistance, ribosome, drug development

Author for correspondence:
Michael Ibba
e-mail: ibba.1@osu.edu

# Translational control of antibiotic resistance

Anne Witzky[1,2], Rodney Tollerson II[2,3] and Michael Ibba[2,3]

[1]Department of Molecular Genetics, [2]Center for RNA Biology, and [3]Department of Microbiology, Ohio State University, Columbus, OH 43210, USA

MI, 0000-0002-5318-1605

Many antibiotics available in the clinic today directly inhibit bacterial translation. Despite the past success of such drugs, their efficacy is diminishing with the spread of antibiotic resistance. Through the use of ribosomal modifications, ribosomal protection proteins, translation elongation factors and mistranslation, many pathogens are able to establish resistance to common therapeutics. However, current efforts in drug discovery are focused on overcoming these obstacles through the modification or discovery of new treatment options. Here, we provide an overview for common mechanisms of resistance to translation-targeting drugs and summarize several important breakthroughs in recent drug development.

## 1. Introduction

Protein synthesis is an essential process that is required by all living organisms. Through the process of translation, the ribosome reads a messenger RNA transcript and synthesizes the encoded protein sequence. Although translation maintains universal importance, there are distinct differences between eukaryotic and prokaryotic translation that have historically been exploited for the development of antibiotics. When a bacterial pathogen is treated with a translation inhibitor, protein synthesis rapidly halts, leading to death or severe growth limitation. Current treatments inhibit translation through a variety of different strategies, ranging from directly targeting the ribosome to targeting aminoacyl tRNA (aa-tRNA) synthetases [1–5]. This method of treating infections has been so effective that dozens of different translation inhibitors have been brought to the clinic since the 1940s, effectively revolutionizing healthcare [6,7]. With the development of antibiotics, an infection that was once life-threatening can now be cleared in a matter of days. However, almost as quickly as these drugs have been developed, they are also being rendered obsolete, as bacteria are rapidly acquiring and evolving mechanisms of resistance in order to avoid eradication. Excessive and inappropriate use of antibiotics has facilitated the rise and spread of multidrug-resistant pathogens, commonly referred to as superbugs [6,8,9]. With the recent plateau in discovery of new classes of antibiotics, infections that are currently treatable may once again become deadly. Despite the looming threat of a post-antibiotic world, efforts to develop new strategies to target translation leave us with an optimistic outlook. Several recent reviews have presented an in-depth look at the mechanism of action of antibiotics targeting translation and specific mutations that confer resistance [1–3,5,10,11]. Here, we bring together the different points in translation that can be targeted for antibiotic development in an overview of common translation drug targets, mechanisms of resistance, and exciting new directions in the development of novel translation-targeting antibiotics.

## 2. Resistance formation

### 2.1. Modification of the ribosome

The ribosome is the central hub of protein production within the cell. Consisting entirely of ribosomal RNA (rRNA) and proteins, the bacterial ribosome is

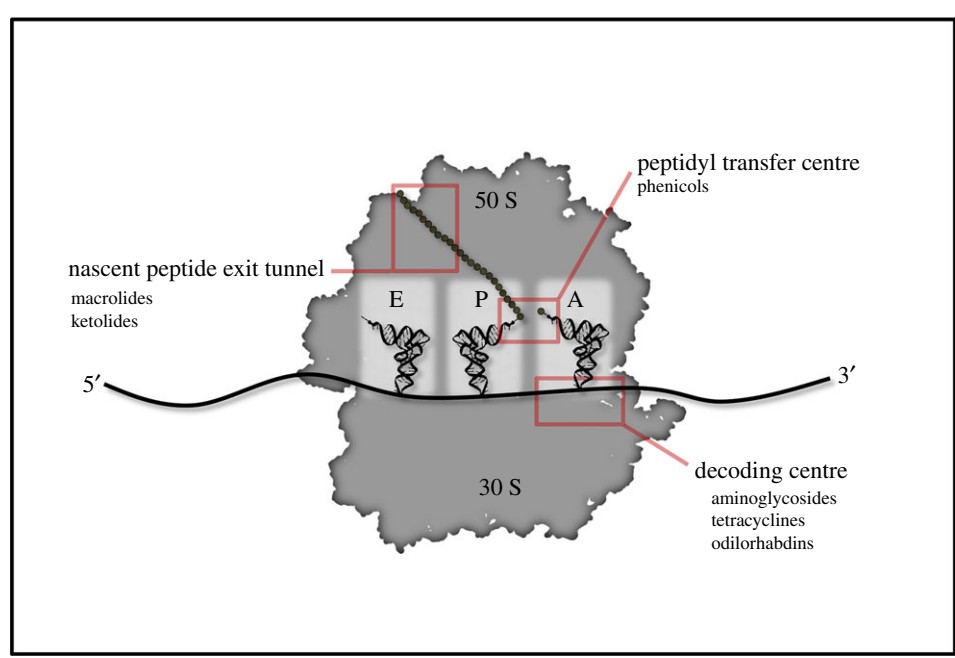

**Figure 1.** General points of inhibition on the 70 S ribosome and discussed drugs that target them.

composed of a large 50 S and small 30 S subunit (figure 1). Due to the large and complex nature of the ribosome, its activity can be inhibited by a variety of different methods with antibiotics that bind in distinct locations [1–3]. For example, aminoglycosides generally target the 30 S subunit, preventing translocation or A-site tRNA binding and promoting miscoding, while macrolides bind the nascent peptide exit tunnel on the 50 S subunit, preventing peptide bond formation and translocation (figure 1) [12,13]. Though both classes of antibiotics maintain distinct activities, both are efficient at halting translation. Modification of the ribosome is one of the most direct forms of antibiotic resistance. Bacteria will often employ methyltransferases to methylate either the large subunit 23 S or small subunit 16 S rRNA [13–15]. Through this methylation, interactions that antibiotics made with either of these rRNAs are prevented, and drug activity is inhibited. Although modification of the ribosome is effective in preventing antibiotic binding, it is not necessarily without its limitations. Some ribosomal modifications alter translation and result in a fitness cost [16]. For this reason, ribosomal modification is often inducible so that it is only used when necessary for survival. For example, erythromycin will induce ribosomal pausing on the leader peptide for the methylase responsible for the dimethylation of the 23 S rRNA, *ermC* [17]. This pausing induces the transcript to form a structure in which the Shine–Dalgarno sequence for *ermC* is exposed, allowing for the translation of *ermC* [15,18,19]. ErmC then methylates the nascent peptide exit tunnel and prevents erythromycin binding. This resistance often comes with a cost, as strains of *Staphylococcus aureus* that are engineered to constitutively express *ermC* are outcompeted by wild-type strains due to the inefficient translation of select polypeptides [16]. This inducible system allows *S. aureus* to survive in the presence of an antibiotic yet still maintain optimal proteome homeostasis when conditions are favourable.

For organisms that maintain numerous copies of rRNA genes, rRNA methylation is an ideal mode of resistance, as each ribosome can receive the modification without the need to acquire the exact same mutation in each rRNA in the genome. However, in organisms that maintain a relatively low number of rRNA genes, copy number is no longer a constraint to the development of resistance mutations. For example, *Mycobacterium tuberculosis* only maintains one copy of each rRNA, and mutation of nucleotides within these rRNAs is an effective way to prevent antibiotic activity. A single amino acid substitution, A1408G, in the 16 S rRNA has been shown to decrease susceptibility of *M. tuberculosis* to aminoglycosides through inhibition of binding [20]. However, as was observed with methylation of the 23 S or 16 S, this mutation comes with a considerable fitness cost. To ameliorate the impact of A1408G, *M. tuberculosis* has been shown to upregulate rRNA methyltransferase *tlyA*, which will then methylate C1409 in the 16 S rRNA [20]. Although this modification does improve the overall fitness of the pathogen, it does lessen the resistance conferred by the A1408G mutation [20]. Through this trade-off, *M. tuberculosis* maintains a middle ground between optimal fitness and effective resistance.

Unlike rRNAs, bacteria generally only maintain a single copy of each ribosomal protein. These proteins can easily acquire mutations to prevent antibiotic activity. For example, mutations in the 30 S ribosomal subunit protein S12 have been shown to induce resistance to miscoding antibiotics such as paromomycin and streptomycin [21–23]. S12 plays a critical role in ensuring the correct codon–anticodon pair is made during decoding. Although identified mutations do not prevent drug binding, they have been shown to significantly increase ribosomal accuracy, effectively counteracting drug activity [24]. *In vitro*, these mutations often also come with a significant fitness cost, as this increase in fidelity also decreases the rate of translation elongation [24,25]. However, second site compensatory mutations can ameliorate this effect, and S12 mutations have also been identified *in vivo* in *M. tuberculosis* that are virtually cost free [26–30]. This suggests that simply reducing antibiotic use will not necessarily result in the elimination of resistant pathogens through competition with the sensitive strains, as is

royalsocietypublishing.org/journal/rsob    Open Biol. 9: 190051

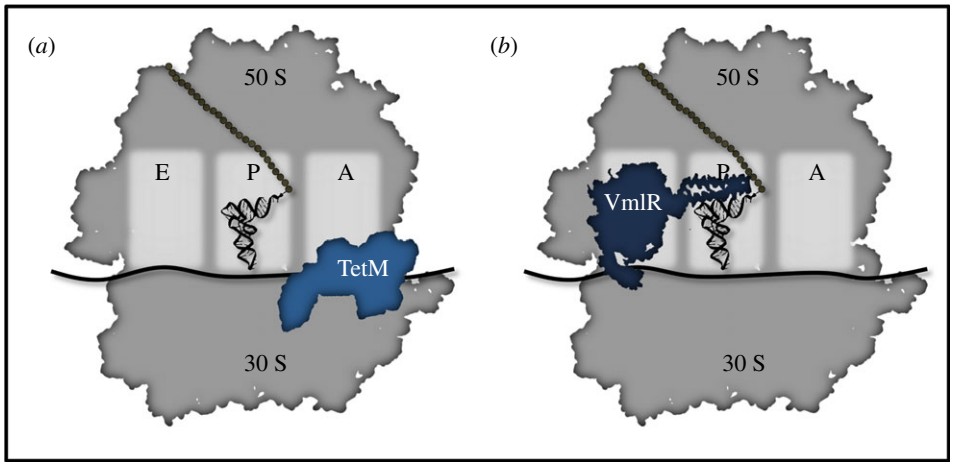

**Figure 2.** Ribosomal protection proteins bind the ribosome to dislodge inhibitory drugs and restore ribosome activity. (*a*) TetM binds the 16 S rRNA in the 30 S subunit to dislodge tetracycline [31]. (*b*) VmlR binds the E site and extends into the peptidyl transfer center to dislodge 50 S targeting drugs such as steptogramin A or lincosamides [32].

commonly believed [26,27]. The gravity of this observation highlights the need for future drug development.

## 2.2. Ribosomal protection proteins

While modification of the ribosome is effective, it generates ribosomes that are not operating at maximum efficiency. As an alternative method of prevention of antibiotic binding, many bacteria employ ribosomal protection proteins (RPPs). An RPP will inhibit drug activity without also permanently altering ribosomal activity. Through direct, reversible binding to the ribosome, several different RPPs have been identified that can prevent drug binding or even dislodge drugs that target either the 50 S or 30 S subunit (figure 2) [11,33].

Both TetM and TetO are RPPs that inhibit the activity of tetracycline, an antibiotic that directly targets the 30 S subunit and prevents aa-tRNA binding at the A site [11,34]. Structural studies have indicated that the protection provided by these proteins is twofold. First, TetM (or TetO) will bind and induce a conformational change in the ribosome. This structural change disrupts key bonds between tetracycline and the ribosome and dislodges it from its binding pocket [31,35]. Second, this new conformation prevents the drug from rebinding, further potentiating the protection [31,35]. Thus, through both of these activities, TetO and TetM are effective in preventing tetracycline binding and removing drug once bound (figure 2*a*). For a recent specialized review on tetracycline resistance mechanisms, see Nguyen *et al.* [11].

A second class of RPPs is the antibiotic resistance (ARE) ATP-binding Casette F (ABCF) proteins. ARE ABCFs are a widespread family of proteins that have been shown to mediate resistance against a diverse set of 50 S targeting drugs. Although they are named for their ability to bind and hydrolyse adenosine triphosphate (ATP), recent evidence indicates they also hydrolyse other nucleotide triphosphates (NTPs) [36]. On average, bacteria maintain four ABCFs per genome, with the most being found in Firmicutes and Actinobacteria [37]. Although the general activity of these proteins had been known for some time, the precise mechanism of action had been controversial. Until recently, there were competing theories postulating that these proteins either associated with a transmembrane domain and acted as efflux pumps, or that they directly inhibited antibiotic activity as RPPs [33]. However, *in vitro* translation experiments provided strong evidence for the latter theory, as addition of an ARE ABCF protein to an *in vitro* translation assay prevented macrolide inhibition of translation in a concentration-dependent manner [38]. This was further solidified with the recent elucidation of the structures of two ARE ABCF proteins, MsrE and VmlR, bound to the 70 S ribosome [32,39]. From these structures, it was evident that these RPPs directly bind to the E-site of the 50 S subunit. Upon binding, the linker domain extends into the peptidyl transfer centre, inducing a conformational change in the ribosome that will result in drug release (figure 2*b*) [32,37,39]. Through these studies, the scope of RPPs was significantly broadened to show that this mechanism of resistance can be applied to the 50 S as well as the 30 S subunit. For a recent specialized review on ABCF-mediated ribosomal protection, see Ero *et al.* [10].

## 2.3. Translation factors

Although the ribosome is the central hub of protein synthesis, it cannot function properly without the assistance of additional translation factors. Translation factor elongation factor G (EF-G) is an essential GTPase that catalyses tRNA translocation through the ribosome. EF-G activity can be directly or indirectly inhibited through the use of several different drugs [40]. For example, fusidic acid binds directly to EF-G and traps it by binding it to the ribosome [41]. Although this will efficiently halt translation, fusidic acid resistance can be easily acquired through point mutations in EF-G that alter drug activity [42]. Such mutations have been identified in clinical isolates of *S. aureus* [42,43]. Although such mutations generally come with a fitness cost, clinical isolates of *S. aureus* have been identified that maintain both the resistance mutation and compensatory mutations to reduce this cost [44]. To lower the rate of resistance formation, fusidic acid is generally only used in a clinical setting in combination with other drugs [43]. Ribosome binding drugs can also indirectly inhibit EF-G activity. Several crystallography studies have found that thiopeptides, GE82832 and dityromycin bind the ribosome and inhibit EF-G-catalysed translocation [3,45]. However, resistance can be acquired to these drugs through point mutations within the ribosomal protein S12 that prevent drug binding [45].

In addition to EF-G, the ribosome also requires another essential GTPase, elongation factor Tu (EF-Tu). EF-Tu binds

aminoacylated tRNAs and escorts them to the A-site of the ribosome. Upon codon–anticodon pairing, EF-Tu will hydrolyse guanosine triphosphate (GTP), release the aa-tRNA and dissociate from the ribosome. Although EF-Tu plays a critical role in translation elongation, drugs found to target EF-Tu generally have low solubility and permeability and are not amenable to clinical use [46]. However, further optimization of such drugs could provide useful compounds in the future. To date, over 30 EF-Tu inhibitors (also known as elfamycins) have been identified [46]. EF-Tu activity can be inhibited with drugs that will trap EF-Tu bound to the ribosome or with drugs that will prevent EF-Tu:GTP aa-tRNA ternary complex formation [47–49]. Although resistance formation against elfamycins has never been characterized in a clinical setting, strides have been made to anticipate potential mechanisms of resistance to such drugs. Given that many bacteria maintain two copies of EF-Tu, elfamycin resistance can be rather complicated. In the case of drugs that prevent ternary complex formation, only one copy of EF-Tu needs to acquire the resistance mutation to continue functioning in translation [50]. In the case of drugs that trap EF-Tu on the ribosome such as kirromycin, both copies of EF-Tu would have to acquire resistance mutations, as a single non-resistant EF-Tu will have a dominant effect due to increased affinity. Resistance to kirromycin has only been observed in bacterial strains in which one copy of EF-Tu has been inactivated [51]. Although this makes elfamycins an enticing lead in drug discovery, significant optimization will be required to improve their pharmacokinetic properties before they are brought to the clinic [46]. For a recent specialized review on elfamycins, see Prezioso *et al.* [46].

## 2.4. Mistranslation-mediated resistance

During translation, the ribosome must correctly pair each codon with the corresponding aminoacylated tRNA so that the appropriate amino acid will be incorporated into the nascent peptide chain. Although accuracy in this process is critical for maintaining the fidelity of the genetic code, errors in translation have been estimated to be as high as $10^{-4}$ per codon [52,53]. These errors in translation are commonly referred to as mistranslation and have been shown to influence antibiotic resistance in a number of different ways.

First, mutations in ribosomal proteins can have a significant impact on ribosome activity, often enhancing or diminishing decoding accuracy depending on the location of the mutation [54]. For example, it has previously been observed that mutations in ribosomal protein RpsD result in a marked increase in the rate of mistranslation [53,55,56]. Although strains harbouring this mutation generally exhibit a decrease in fitness due to the high number of errors in their proteome, these errors can be advantageous for the population as a whole when confronted with cefotaxime [54]. In the presence of low levels of this drug, weakly deleterious mutations in the cefotaxime resistance protein (TEM-1) are tolerated (figure 3). Although such a mutation would not be beneficial, when the concentration of the antibiotic is low, the fitness cost associated with the mutation is not significant enough to result in complete elimination of the organism. However, if the organism maintains a high level of mistranslation, then the TEM-1 can harbour both this weakly deleterious mutation as well as mistranslated residues. This combination exacerbates the deleterious effects

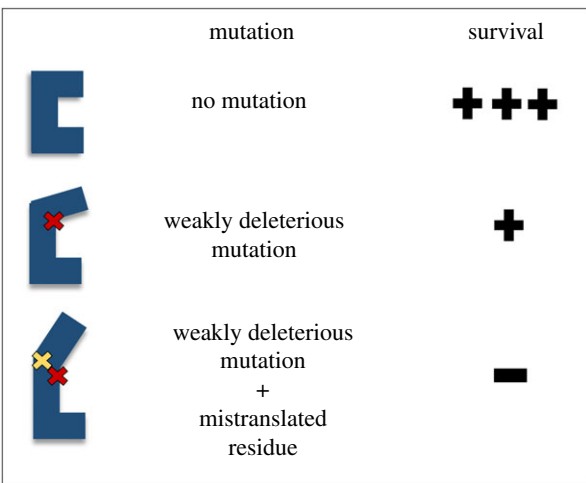

**Figure 3.** Mistranslation exacerbates deleterious mutations in antibiotic-resistance factors. An antibiotic-resistance protein (blue) with mutation (red X) can maintain partial activity, and bacteria with such a mutation survive in the presence of low levels of antibiotic. Activity is lost when protein maintains both the mutation and an additional mistranslated residue (yellow X), and the bacteria die in the presence of low levels of antibiotic.

of the mutation, and it is effectively purged from the population, resulting in enhanced fitness overall (figure 3) [54]. In this way, mistranslation can be advantageous to a population that is met with an antibiotic stress. However, it is noteworthy that this effect is only observed in the presence of low levels of antibiotic, where organisms with weakly deleterious mutations are able to survive. When the antibiotic stress is high enough, even weakly deleterious mutations in an antibiotic-resistance gene cannot be tolerated. Although this suggests that this phenomenon would not be relevant in a patient being treated with a high dose of an antibiotic, it could be critical for the survival of resistant pathogens in the environment. With the inappropriate and overuse of antibiotics, low levels of these drugs make their way into the environment [6]. Within this context, it is likely that mistranslation could aid in enhancing fitness of organisms that are tolerating this low level of antibiotic stress.

A second source of mistranslation arises from errors in aminoacylation of tRNAs. Aminoacyl tRNA synthetases (aaRSs) are the enzymes responsible for correctly pairing each tRNA to its cognate amino acid. Errors in this process can result in misacylated tRNAs that are then used for translation, allowing non-cognate amino acids to be incorporated into a protein. Mutation of aaRSs and related proteins can disrupt the fidelity of aminoacylation and allow for mistranslation. Such mutations have often been found in clinical isolates that display enhanced antibiotic resistance. For example, clinical isolates of *M. tuberculosis* have been shown to harbour mutations in *gatA*, an amidotransferase that facilitates the conversion of Glu-tRNA$^{Gln}$ to Gln-tRNA$^{Gln}$ and Asp-tRNA$^{Asn}$ to Asn-tRNA$^{Asn}$ [57]. These mutations in *gatA* result in an increase in misincorporation of Glu at Gln codons and Asp at Asn codons. Through such misincorporation, an Asn residue in RpoB that is critical for rifampicin binding is mistranslated as an Asp. This substitution prevents rifampicin binding and makes *M. tuberculosis* highly tolerant of rifampicin treatment [57,58].

Mutations in aaRSs have also been shown to influence antibiotic resistance through indirect effects. For example, strains that are resistant to ciprofloxacin often harbour

royalsocietypublishing.org/journal/rsob    Open Biol. 9: 190051

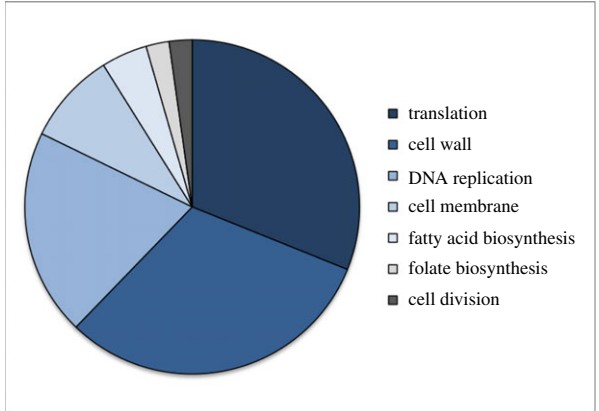

**Figure 4.** Cellular targets of antibiotics currently under clinical development based on analysis from the PEW Charitable Trusts [64].

- translation
- cell wall
- DNA replication
- cell membrane
- fatty acid biosynthesis
- folate biosynthesis
- cell division

mutations in several different aaRSs [59]. These mutations result in inappropriate activation of the stringent response, presumably through the accumulation of deacylated tRNAs. Through the stringent response, the cell activates a number of efflux pumps, which remove the antibiotic from the cell. Although these mutations were initially identified in the presence of ciprofloxacin, activation of the stringent response through aaRS inhibition was also shown to increase resistance to drugs that act through diverse mechanisms including rifampicin, chloramphenicol, mecillinam, ampicillin and trimethoprim, indicating that partial aaRS inactivation can have broad implications for resistance formation [59]. *Escherichia coli* clinical isolates have also been identified that harbour similar aaRS mutations, indicating that this mechanism is likely relevant *in vivo* [60–63].

# 3. New frontiers in antibiotic development

Mechanisms of resistance have been identified for nearly every translation inhibitor in the clinic [6]. As many of these drugs are being inappropriately overused, resistance will likely continue to spread until these drugs become obsolete [6,8,9]. This ominous threat merits the development of new treatment options. Given the past success of translation inhibitors as therapeutics, many are developing new strategies to target this process. Currently, approximately one-third of new drugs under clinical development target translation (figure 4) [64].

## 3.1. Discovery of novel antibiotics

The years between 1940 and 1960 were the golden age of antibiotic development [6]. During this time, the majority of the antibiotics used today were identified in soil-dwelling actinomycetes. However, as only 1% of bacteria can be cultured in a laboratory setting, the number of compounds identified from these organisms is limited [65]. After this short 20-year span, screening cultivable actinomycetes for drugs was only leading to rediscovery of known compounds. This roadblock was a major factor behind the plateau in antibiotic development. However, in recent years, scientists have begun to revisit this approach. Within previously unculturable organisms, several promising new antibiotics have been identified.

Actinomycetes maintain a high number of non-ribosomal peptide synthetases (NRPSs) and polyketide synthetases (PKSs) that allow for the production of diverse metabolites

and antibiotics. Due to the previous success in identifying drugs in actinomycetes, it is likely that other organisms that maintain genes with similar functions will also produce useful compounds. Like actinomycetes, the *Xenorhabdus* genus maintains a relatively high number of diverse NRPSs and PKSs [66]. However, *Xenorhabdus* is not typically used for antibiotic discovery, as it is naturally a symbiont in soil-dwelling nematodes. In a recent screen for antimicrobial activity in *Xenorhabdus* extracts, a novel class of translation-targeting antibiotics was identified. Odilorhabdin binds the 30 S ribosomal subunit at the decoding centre and induces ribosomal stalling and miscoding, presumably through increasing affinity of non-cognate aa-tRNAs for the A-site [67]. Though this activity is reminiscent of aminoglycoside, tetracycline and negamycin inhibition, the binding site of odilorhabdin does not overlap with any of these drugs [67]. Further, odilorhabdin displays antimicrobial activity for a wide spectrum of both Gram-positive and Gram-negative pathogens while exhibiting low levels of toxicity in a murine model, indicating odilorhabdin has a promising future as the basis for new therapeutics [67].

Attempts have also been made to identify drugs in the other 99% of bacteria that have never been cultured. Through the use of isolation chips (iChips), scientists can now culture bacteria directly in a soil environment, allowing for the identification and characterization of previously uncultured organisms [68]. New antibiotics can then be identified in the extracts of these elusive organisms. For example, through the use of iChips, teixobactin, a cell wall biosynthesis inhibitor, was identified in *Eleftheria terrae*, an organism that was previously uncultivable [69]. Although teixobactin does not inhibit translation, this discovery highlights the potential for discovery of novel translation-targeting drugs in previously uncultured organisms.

## 3.2. Modification of existing antibiotics

Although the isolation of antibiotics from natural sources is a relatively new development in modern medicine, these drugs and the corresponding resistance genes are ancient [70–72]. Over millions of years, bacteria have evolved these compounds and protection mechanisms for competition, communication and regulation of gene expression. Isolation of previously existing natural compounds comes with both significant benefit and cost. On the one hand, the drugs that are being isolated are the result of millions of years of evolution and are therefore highly effective. On the other hand, the genes that are required for resistance have also been selected for over evolutionary time and can easily spread between organisms in the presence of sufficient selective pressure. In an effort to still use these potent antibiotics while avoiding resistance mechanisms, drug development is turning to the chemical modification of previously isolated compounds.

The modification of tetracycline has resulted in several aminomethylcyclines and fluorocyclines that are currently under clinical development or that have recently been approved by the US Food and Drug Administration (FDA) [64]. The aminomethylcycline omadacycline was FDA approved in 2018 to treat community-acquired bacterial pneumonia and acute bacterial skin and skin structure infections and KBP-7072 (another aminomethylcycline) is in clinical development for the treatment of pneumonia and is effective against *S. aureus* and *Streptococcus pneumoniae* in a murine model [64,73]. Of the

fluorocyclines, TP-271, TP-6076 and eravacycline have displayed potent *in vivo* efficacy, and eravacycline was FDA approved in 2018 [64,74–78]. These drugs have also been shown to be effective *in vitro* against tetracycline-resistant clinical isolates, including isolates that are known to maintain ribosomal protection proteins [76,78]. This indicates that these modified drugs have the potential to avoid existing resistance mechanisms.

Aminoglycosides have also been used as a scaffold for chemical modification to generate more powerful antibiotics. Plazomicin (ACHN-490) is a modified aminoglycoside that was FDA approved in 2018 [64,79]. It is effective against drug-resistant clinical isolates, including strains that are known to be resistant to aminogyclosides and carabapenem-resistant Enterobacteriaceae, which have been identified by the Centers for Disease Control as an urgent threat in the rise of antibiotic-resistant pathogens [79,80]. However, plazomicin was not effective *in vitro* against strains expressing 16 S methylase armA, indicating that it is not immune to all modes of aminoglycoside resistance [79]. Attempts have also been made to use aminoglycosides as carriers that will target a 'catalytic warhead' directly to the ribosome. Neomycin can be modified with diverse diamine compounds that have been shown to accelerate the cleavage of adenylyl(3′–5′)adenosine [81]. It would be expected that, as this modified neomycin binds the ribosome, the diamine would stimulate cleavage of the 16 S rRNA, permanently disabling the ribosome. Although initial results do not reveal rRNA cleavage, structural data indicate that two diamine modifications do induce a significant structural change in the phosphate backbone of the 16 S rRNA [81]. Though this change is not sufficient to stimulate rRNA cleavage, this result indicates that, through remodelling of the diamine modification, these warheads will be effective in disabling ribosomes.

Synthetically derived ketolides are a promising class of antibiotics that are currently under clinical development [64,82,83]. Nafithromycin (WCK 4873) and solithromycin (T-4288) are effective against a wide range of pathogens, including some that are known to be resistant to macrolides [84–86]. Although a recent metanalysis indicated that such ketolides are not necessarily more potent than current treatments, these drugs are believed to not fully induce expression of the corresponding resistance gene and are therefore powerful alternatives to avoid this mechanism [84,87]. That being said, such drugs are less effective against strains that display a constitutive macrolide–lincosamide–streptogramin B-resistance phenotype [84,86]. It is important to also note that ketolides could come with additional limitations. Although all of the clinically relevant ketolides are chemically derived from macrolides, a single naturally occurring ketolide, pikromycin, has been isolated from *Streptomyces venezuelae* [88]. In order to resist inhibition from the pikromycin it produces, *S. venezuelae* must also maintain two inducible methyltransferases (*pikR1* and *pikR2*) that provide resistance to this drug. Despite the diversity of the synthetically derived ketolides, bacteria that have acquired *pikR1* or *pikR2* become resistant to the synthetic compounds, suggesting that chemically derived compounds are not immune to all resistance mechanisms [88].

Recently, strides have also been made in the modification of chloramphenicol, a phenicol that binds to the petidyl transferase centre and inhibits accommodation of the A-site aa-tRNA

(figure 1) [89,90]. Aminoacyl derivatives of chloramphenicol have been shown to display both an increased affinity for the ribosome and an altered drug binding site [89]. Owing to this change in site specificity, structural data suggest that these derivatives would not be inhibited by methylation of the 23 S rRNA, a common resistance mechanism against phenicols. However, these analogues do not display the same level of translation inhibition as the parent molecule [89]. Further chemical optimization would be required for these modified compounds to be useful in a clinical setting. Derivatization of chloramphenicol can also have a significant impact on which cellular process the drug will target. For example, chloramphenicol-derived enone and enal analogues have been shown to inhibit cell wall biosynthesis rather than translation [90]. Although these derivatives have a drastically different mechanism of action, they are still highly efficient in inhibiting bacterial growth and exhibit a significantly lower propensity for resistance formation than the parent molecule. Despite these advantages, these analogues will also require more optimization before they are ready for a clinical setting, as they display high levels of toxicity in mammalian cell lines [90].

Many groups are taking the chemical modification of drugs a step further by creating hybrid compounds that merge two existing drugs with distinct mechanisms of action. For example, MCB3837 (DNV3837) is an oxazolidinone/quinolone hybrid that is under clinical development for the treatment of *Clostridium difficile* [64]. MCB3837 is converted to MCB3681 upon intravaneous infusion, and MCB3681 has been shown *in vitro* to have a lower minimum inhibitory concentration (MIC) than cadazolid, fidaxomicin, metronidazole and vancomycin against *C. difficile* clinical isolates [91]. As *C. difficile* is an opportunistic pathogen that colonizes as other bacteria are depleted, significant efforts have been made to analyse changes in the microbiome in response to MCB3837 treatment [92]. In proof-of-principle human trials, MCB3837 had no significant impact on the levels of resident Gram-negative bacteria in the human skin, nose, oropharynx and intenstine microbiome, but there was a significant reduction in clostridia, bifidobacteria, lactobacilli, enterococci and *S. aureus* levels, indicating that MC3837 does alter Gram-positive colonization, and could be effective against *C. difficile* in the clinic [93,94].

## 3.3. Using translation for developing adjuvants

As a considerable amount of time and resources have gone into the development of drugs currently in the clinic, it would be desirable to prevent these treatments from becoming obsolete. An alternative to complete drug redesign is to identify compounds that can be used to potentiate current treatments. Such additive therapies are referred to as adjuvants, and they present several advantages in the future of drug development. Adjuvants have the potential to broaden the use of existing drugs and directly overcome resistance mechanisms. As many adjuvant therapies in development do not impact viability, they are often able to achieve this without generating significant selective pressure for resistance formation [95]. Several new strategies are under development to look for adjuvants that will potentiate translation inhibitors.

One strategy in adjuvant development is to identify compounds that will make existing drugs more effective against a broader spectrum of bacteria. Many large, hydrophobic translation-targeting drugs are ineffective against Gram-

royalsocietypublishing.org/journal/rsob    Open Biol. 9: 190051

royalsocietypublishing.org/journal/rsob Open Biol. 9: 190051

negative organisms, as they maintain an additional outer membrane that prevents penetration of such drugs [96–99]. This barrier has resulted in fewer treatments available for Gram-negative infections [65,96]. Many groups have used high-throughput screening to look to adjuvant treatments that sensitize the Gram-negative outer membrane to broaden the scope of available treatments for such infections. Polymyxin B non-apeptide (PMBN), pentamidine and oligo-acyl-lysyls (OAKs) are small molecules that permeabilize the Gram-negative outer membrane and allow for more efficient penetration of translation inhibitors that are currently only used to treat Gram-positive infections [96,100–103]. Although these treatments are not available in the clinic, they have shown promise *in vitro* against clinical isolates and in murine and *Galleria mellonella* infection models [96,100–103]. For example, PMBN with erythromycin and OAK with rifampicin combination therapies improve mouse survival after *Klebsiella pneumoniae* infection [100,103]. Such adjuvants have the potential to broaden the scope of pathogens that can be cleared with drugs that are already FDA approved [102].

An alternative strategy in adjuvant development is to identify drugs that directly inhibit resistance factors. For example, resistance to aminoglycosides is often conferred by aminoglycoside-modifying enzymes (AMEs) that inactivate aminoglycosides with the addition of an acetylation or phosphorylation [104]. Several groups have made strides towards identifying AME inhibitors. Aminoglycoside bisubstrates and cationic antimicrobial peptides have been shown to directly inhibit AMEs *in vitro*, and several aminoglycoside bisubstrates also have been shown to potentiate kanamycin against *Enterococcus faecium* [105–108]. Inhibitors of the acetyltransferase Eis (enhanced intracellular survival) also potentiate kanamycin against *M. tuberculosis* [109]. In a similar strategy, enzymes that modify the ribosome can also be directly targeted for adjuvant development. Several groups have designed *in silico* or *in vitro* high-throughput screens to identify compounds that target 23 S methylase ErmC and prevent interactions with either the substrate RNA or S-adenosyl-L-methionine [110–112]. *E. coli* expressing ErmC are sensitized to erythromycin when co-treated with such compounds [110,111]. ErmC inhibitors also lower the azithromycin MIC for *S. aureus*, *E. coli* and *Enterococcus faecalis in vitro*, but this result was not supported in a murine model [112].

Pathogens have also been shown to use mistranslation as a mode of resistance by altering target residues and prevent antibiotic binding. In one of the aforementioned examples, *M. tuberculosis* relies on the mistranslation of RpoB to prevent rifampicin binding [57,58]. For this reason, preventing mistranslation in *M. tuberculosis* is an attractive target for developing an adjuvant therapy to potentiate rifampicin treatment. As kasugamycin has previously been shown to increase ribosomal accuracy, it is an intriguing candidate for such a therapy. It has been demonstrated that kasugamycin does in fact increase susceptibility of mycobacteria to rifampicin both *in vitro* and *in vivo* [113]. However, the combination of the drugs is poorly tolerated in a murine model system and therefore unlikely to be a viable option in a clinical setting [113]. Although kasugamycin specifically may not be appropriate for patients, this example highlights the potential for new drugs that alter translational accuracy to be used as adjuvant therapies.

Bacterial translation can also be directly targeted in adjuvant development. Elongation factor P (EF-P) is a universally conserved translation factor that is required for efficient translation of polyproline motifs [114,115]. Although EF-P has the same functional role in all bacteria characterized thus far, the relative importance of EF-P in maintaining cellular physiology varies between different organisms [116–123]. Despite the clear differences in the physiological significance of EF-P, one unifying feature remains: antibiotic sensitivity. Although it does not directly inhibit drug activity, EF-P has repeatedly been shown to establish antibiotic resistance in diverse pathogenic and non-pathogenic bacteria [119,121,123]. It is noteworthy that *efp* mutants are hypersensitive to drugs with diverse mechanisms of action. For example, a *Salmonella enterica efp* mutant displays increased susceptibility to polymyxin B, a cell wall biosynthesis inhibitor, as well as gentamicin, an aminoglycoside targeting the 30 S subunit of the ribosome [119]. It is also important to note that, although EF-P is required to maintain proteome homeostasis and full antibiotic resistance, this impact is limited to conditions of rapid growth. Recent evidence indicates that EF-P activity is dispensable under conditions that induce slow growth such as low temperatures and nutrient limitation, presumably due to decreased translational demands [124]. Therefore, the antibiotic hypersusceptibility observed in an *efp* mutant is abolished when growth of the pathogen is slowed. Given that a pathogen must undergo rapid proteomic reprogramming as it adapts to survive in various cell types within a host, within the context of infection, the role of EF-P in facilitating antibiotic resistance is highly relevant [125]. Though, it would likely lose this relevance in the case of dormant persistors. This variability in the requirement for EF-P highlights the potential for culture conditions to influence our interpretation of resistance formation and drug discovery. For any infection, the pathogen is not present as a pure culture, but rather a single player within the vast human microbiome. Such different environmental conditions can have a significant impact on the relevance of any one translation factor and should be considered in future work.

## 4. Conclusion

Bacterial translation has been successfully exploited for the development of powerful antibiotics. Though they maintain diverse mechanisms of action, translation inhibitors have been used in the treatment of a wide variety of infections and saved countless lives. However, with the rise and spread of antibiotic-resistant pathogens, these treatments are being rendered obsolete [6,7]. This threat is rapidly becoming a global health crisis, resulting in tens of thousands of deaths and billions of dollars in added healthcare costs. Although scientists have made significant strides in the fight against antibiotic resistance with translation inhibitors, targeting translation comes with limitations. Translation is an essential process, and inhibition of any process required for viability creates immense selective pressure. Only pathogens that have evolved or acquired resistance are able to survive, and they are able to thrive as non-resistant competitors are eliminated. Furthermore, as translation is equally important for the host's beneficial microbiome, translation inhibitors can have undesirable side effects. Development of new drugs that do not impose such a strong selective

pressure will be a significant challenge. An additional problem in the future of drug development will be overcoming persistence, a dormancy state that allows bacteria to survive in the presence of an antibiotic and then resume growth after treatment. Persistence is particularly problematic in a clinical setting, as it has been shown to increase the likelihood of antibiotic-resistance formation and translation inhibitors are ineffective against such pathogens [126]. Despite these challenges in drug development, translation inhibitors have proven to be effective therapeutics in many settings. With new strategies in antibiotic identification and redesign, translation may yet be the perfect solution to antibiotic resistance.

Data accessibility. This article has no additional data.

Competing interests. Anne Witzky is the co-founder of Avirent, LLC.

Funding. We received no funding for this study.

Acknowledgements. We thank Rebecca E. Steiner and Paul M. Kelly for helpful discussions and critiques and Katherine A. Fairman for assistance in generating figures.

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
