## [Reviewer comments · Open Biology]

Review History

RSOB-19-0051.R0 (Original submission)

Review form: Reviewer 1

Recommendation

Major revision is needed (please make suggestions in comments)

Are each of the following suitable for general readers?

- a) **Title**
Yes
- b) **Summary**
Yes
- c) **Introduction**
Yes

Is the length of the paper justified?

Yes

Should the paper be seen by a specialist statistical reviewer?

No

Is it clear how to make all supporting data available?

Not Applicable

Is the supplementary material necessary; and if so is it adequate and clear?

Not Applicable

Do you have any ethical concerns with this paper?

No

Comments to the Author

This review address two complementary subjects: One is to review mechanisms by which protein synthesis and translation can be protected from inhibition by existing antibiotics. The second is to survey 'exciting new directions' in the development of antibiotics targeting translation and protein synthesis.

General comments.

I think the review is rather short, and does not seem especially comprehensive. I could imagine another review that for example goes more comprehensively into all of the tet RPP genes, the various specific mutations in ribosomal proteins conferring resistance to various drugs, and all of the RMT genes and which nucleotides they modify and the effects that they have. In addition, I think it is important to consider the fact that many translation-inhibiting antibiotics are limited in their clinical use by their spectrum of antibacterial activity - some are ineffective against Gram-negatives while others have a very broad spectrum. Compared to my hypothetical review, this review seems superficial. On the other hand, it functions as a quick, broad survey of the field of resistance related to translation and protein synthesis.

At times I think it loses track of what it is supposed to be discussing in each section as well. The antifungal tRNA synthetase drug section in particular seems like it doesn't fit where it is. Likewise the long discussion of EF-P (pages 8-9) is at best of marginal interest to clinical antibiotics and resistance and could be greatly shortened or cut completely. Similarly for the long discussion of rpoD and mistranslation.

I get the impression that the authors have placed an artificial limitation on themselves to only discuss one or two examples of each phenomenon, which may be why the review often seems superficial. The section where this is a serious problem is in the New Frontiers in Antibiotic Development section where I don't think they can have this section with only odorhabdins as an example. The Pew Trusts database maintains a listing of all antibiotics in clinical trials development globally (current and past decade at least). I think the authors should consult this database and include in their review information on the multiple new aminoglycosides, tetracyclines, macrolides, oxazolidinones, etc that are being brought through clinical trials or have recently been approved for clinical use. For example, one interesting compound is a chemical hybrid of an oxazolidinone and quinolone, dual targeting translation and topoisomerase. This compound went all the way through Phase 3 clinical trials but development has recently been suspended by the company. Nevertheless, I think it is important to bring this hybrid up as an important example of a direction in which translation targeting antibiotics could go. It would also be important to discuss why new variants of classical classes are being developed - what problems are they addressing (in many cases it is connected with overcoming resistance mechanisms and so should be of interest for this review).

Here are some line comments:

Line 56: Does halting protein synthesis rapidly lead to elimination of bacteria? Chloramphenicol and Macrolides halt translation and are bacteriostatic.

Line 77 (Figure 1). Figure 1 serves to give a very general overview of important ribosomal locations for antibiotic action. I think however, that the authors should consider including either additional more detailed figures (or additional panels to this figure) showing the locations of interaction of several specific classes of antibiotics, including the odilorhabdins.

Line 116. It is not correct to state that all S12 mutations come with a significant fitness cost. Multiple publications have shown in various species, and in animal infections, that at least one S12 mutations is essentially cost-free. As an example I suggest you include reference to a paper by Erik Böttgers group showing a strong bias in favour of this mutation in resistant *Mycobacterium tuberculosis* clinical isolates: Böttger et al, (1998) *Nature Medicine* 4, 1343-1344.

Line 129: As I read them, both of the citations propose or support models that TetO and TetM dislodge bound tetracycline and induce a conformational change that prevents tetracycline rebinding after the RPP has left. I did not see anything that indicated evidence for direct competition for ribosomal binding between RPPs and tetracycline. I think you should rewrite this section to better reflect the experimental evidence.

Line 135: ABCF needs to be defined and introduced for the uninitiated. It seems strange to introduce them with their most important characteristic being that they are NTP hydrolases.

Line 159 - 160. I think you should give credit to the references with precedent for fusidic acid resistance mutations in *S. aureus*. I suggest you include reference to Nagaev et al, (2001) *Mol Microbiol*, 40, 433-439; and Laurberg et al (2000) *J Mol Biol* 303, 593-603.

Lines 186-219. I agree that EF-P is an interesting and important translation factor. However, it is not a target for antibiotics and I think this section could be entirely removed (or reduced to a very short comment on its potential as a future target).

Lines 221-248. Similarly to the above, mistranslation is interesting for ribosomologists but it is not a significant player in antibiotic therapy and this section should either be removed or reduced to a very short commentary on its potential for use.

Lines 236-239: In addition: This is very unclear to me and needs to be reworded to make the intended meaning clear. I think that some of the unclearness may come from the introduction of an 'antibiotic resistance protein' in the middle of this section.

Lines 272-280: This section about drugs targeting tRNA synthetases is out of place in a discussion of resistance mechanisms. It should be moved to the New Frontiers in Antibiotic Development section. I think that tRNA synthetases as novel drug targets should be discussed but you must be more comprehensive and include the examples of GSK2251052 and the recent GSK Tuberculosis tRNA synthetase inhibitors. You should also mention the one that has been on the market for decades - mupirocin.

The New Frontiers in Antibiotic Development section should be expanded. There have been other developments in translation targeting drugs other than odilorhabdins. Please examine the literature and especially the Pew Trust database (there are several aminoglycosides and tetracyclines and macrolides designed to overcome resistance mechanisms, for example).

Lines 331 - 346. It seems wrong to have this long speculation about modification of chloramphenicol while ignoring the actual aminoglycosides, tetracyclines, macrolides etc that are in clinical trials or recently approved.

Lines 347 - 355. Similar criticism to above - this text on modifications of neomycin is speculative ribosomology rather than a discussion of developments in antibiotic development. In a long review there would be a place for such speculations but not at the expense of ignoring all of the developments that are actually occurring.

Lines 356 - 365. Same comments as above - you ignore the actual macrolides and ketolides in development and instead return to discussing a potential resistance mechanism.

Lines 366 - 394. This section is very speculative and it would be better to devote text to actual examples where an adjuvant has been shown to increase the potency of an antibiotic. There are publications where adjuvants have for example been used to increase the entry of Gram-positive antibiotic (macrolides) into Gram-negatives. These have not made it to the clinical yet but I think it would be appropriate to discuss the potential of actual examples like this rather than purely speculative examples (like mistranslation/kasugomycin and EttA ADP/ATP ratios).

Lines 409 - 416. I don't think the discussion of persisters is relevant to the review and certainly is not a good way to end the review. Better to end with a summary of the major developments in drug discovery in the area and the problems they are trying to address.

Review form: Reviewer 2

Recommendation

Major revision is needed (please make suggestions in comments)

Are each of the following suitable for general readers?

- a) **Title**
Yes
- b) **Summary**
Yes
- c) **Introduction**
Yes

Is the length of the paper justified?

Yes

Should the paper be seen by a specialist statistical reviewer?

No

Is it clear how to make all supporting data available?

Not Applicable

Is the supplementary material necessary; and if so is it adequate and clear?

Not Applicable

Do you have any ethical concerns with this paper?

No

Comments to the Author

This review by Witzky et al. provides an overview of current knowledge of bacterial resistance mechanisms against antibiotics that target the ribosome (and other components of the translational machinery), along with discussion of work towards or new potential strategies that might alleviate this important clinical problem. This significance of this topic is clearly high – antibiotics that target protein synthesis have been critical in treatment of bacterial infection but many (if not all) are threatened by one or more forms of resistance. The global threat of antibiotic resistance is widely acknowledged, making this review of interest to a potentially wide audience. As acknowledged by the authors, given the importance of these drugs and the topic in general, several other detailed reviews have been published in recent years that cover topics of translation-targeting antibiotics and their interactions with the ribosome, mechanism of action and resistance mechanisms. However, the approach taken by the authors here to overview resistance mechanisms together with substantial discussion of future perspectives on the opportunities and challenges faced in overcoming this resistance gives the work a unique perspective and makes it potentially a valuable contribution to this field.

While I am overall positive about the potential contribution of this work, to maximize its impact, I encourage the authors carefully review a number of aspects of how the material is presented. Three main aspects could be strengthened:

Eliminating a number of instances of vague or overly informal language, oversimplification of important concepts, and the odd example of hyperbole (do bacteria really “run rampant?”). While it is important for a review of this nature to be broadly accessible this should not come at the expense of an appropriate level of accuracy and rigor (some examples are included in the specific comments below).

A single, not especially useful figure is used to illustrate the entire discussion. This figure should feature earlier and could be substantially improved e.g. label binding sites of the antibiotics and resistance mechanisms discussed (e.g. where modifications or mutation lead to resistance). All elements of the figure should be labeled (not just the subunits-mRNA, tRNA, nascent peptide). A second panel summarizing the discussion of aaRS targeting antibiotics and resistance mechanism would also be useful. More generally, many other aspects of the discussion would benefit from illustrations – these could be structures of proteins (such as ribosome bound protection factors) involved highlighting key features mentioned in the text, model figures summarizing mechanism of action, etc.

In the section on adjuvant strategies (or in its own sub-section) the approach of directly inhibiting resistance determinants (“resistance breakers”) should be included.

Specific comments and suggestions:

1. Line 56, “protein synthesis rapidly halts, leading to elimination of the bug” ... how does the drug eliminate? Bacteriostatic vs bactericidal activity, etc. (I also generally object to the use of the term “bug”!)
2. Line 63, “bacteria rapidly evolving mechanisms of resistance” ... not clear this entirely true; resistance mechanisms can be acquired by gene transfer and also (as later noted) there is compelling evidence for the ancient nature of resistance (pre-human use of antibiotics).

3. Line 75, specify “bacterial ribosome” (others are not 30S/50S)
4. Line 77,78 – include inducing miscoding as mechanism of action of aminoglycosides (this is mentioned later but should be stated here)
5. Line 82, “Bacteria will often employ inducible methyltransferases”... this is not so clear cut and could be described in a little more detail (how are they inducible, under what circumstances; drug producer vs. acquired resistance enzyme in a pathogen, etc)
6. Line 86-87, statement that ribosome modification “often” creates a significant impediment should be supported by references. Some elaboration may also be needed, e.g. it is not so clear cut that this is the case for all aminoglycoside resistance.
7. Line 133, grammatical issue? (suggest “preventing tetracycline binding and removing drug once bound”)
8. Line 156, “directly OR indirectly”
9. Line 165, mutations in rRNA, protein, both?
10. Line 168, this statement is misleading – EF-Tu hydrolyses GTP regardless of whether correct codon-anticodon pairing occurs.
11. Line 181, ribosomes are not “permanently” inactivated – competitive inhibitor with a dominant effect due to increased affinity.
12. Lines 212-220 (discussion of EF-P). Not really clear what the main take-away point is. here. What is the evidence for high translational demand? Does need to evade immune system or outcompete commensals depend on location/ type of infection/ pathogen? What about dormancy (also an effective strategy to evade antibiotic treatment, also immune system)? Do common commensals have EF-P and should this also be a consideration here?
13. Lines 229-239, I also found this description of RpsD mediated effects confusing. In line 236 why “also” maintains? (Isn’t the discussion about RpsD leading to mistranslation?) And, which “antibiotic resistance protein” is referred to in Line 237?
14. Line 252-253, “non-cognate” is by definition in “the incorrect location”.
15. Line 323, while likely true at least in part, interspecies competition is probably not the only reason for bacterial evolution of antibiotics (intra/ interspecies “communication”, regulation of gene expression, etc). The wording here is also a little confusing in terms of natural vs. semi-synthetic antibiotics: Line 324 appears refer to alteration of existing (natural) drugs; additionally, there is no “this method” described in the preceding sentence and also the next sentence refers to them as being “isolated” (which suggests the natural compounds themselves). Finally, the paragraph ends with the statement that efforts have turned to chemical modification (which is where it seemed start...).
16. Line 332, first reference to the figure (despite multiple prior mentions of tRNA binding sites or other features).
17. Line 338, instability of what? The drug itself? (Chemical lability, greater sensitivity to degradation). Is it shown that the alternative site isn’t just less effective for ribosome inhibition?

18. Line 339, "Derivatization of chloramphenicol can also... impact drug activity." Isn't this also the case for the previous example? If so, why "also"? (It's not completely clear but I assume here "activity" refers to target or mechanism of action? However, even that changed in the previous example as the ribosome binding site was altered).

19. Phrase "Translation is an essential process" is repeated in Line 396 & 403.

Decision letter (RSOB-19-0051.R0)

01-Apr-2019

Dear Dr Ibba:

We are writing to inform you that the Editor has reached a decision on your manuscript RSOB-19-0051 entitled "Translational control of antibiotic resistance", submitted to Open Biology.

As you will see from the reviewers' comments below, there are a number of criticisms that prevent us from accepting your manuscript at this stage. The reviewers suggest, however, that a revised version could be acceptable, if you are able to address their concerns. If you think that you can deal satisfactorily with the reviewer's suggestions, we would be pleased to consider a revised manuscript.

The revision will be re-reviewed, where possible, by the original referees. As such, please submit the revised version of your manuscript within six weeks. If you do not think you will be able to meet this date please let us know immediately.

When submitting your revised manuscript, please respond to the comments made by the referee(s) and upload a file "Response to Referees" in "Section 6 - File Upload". You can use this to document any changes you make to the original manuscript. In order to expedite the processing of the revised manuscript, please be as specific as possible in your response to the referee(s).

Please see our detailed instructions for revision requirements
<https://royalsociety.org/journals/authors/author-guidelines/>

Sincerely,
The Open Biology Team
<mailto:openbiology@royalsociety.org>

Editor's Comments to Author(s):

Please address all comments of the reviewers

Reviewer(s)' Comments to Author(s):

Referee: 1

Comments to the Author(s)

This review address two complementary subjects: One is to review mechanisms by which protein synthesis and translation can be protected from inhibition by existing antibiotics. The second is to survey 'exciting new directions' in the development of antibiotics targeting translation and protein synthesis.

General comments.

I think the review is rather short, and does not seem especially comprehensive. I could imagine another review that for example goes more comprehensively into all of the tet RPP genes, the various specific mutations in ribosomal proteins conferring resistance to various drugs, and all of the RMT genes and which nucleotides they modify and the effects that they have. In addition, I think it is important to consider the fact that many translation-inhibiting antibiotics are limited in their clinical use by their spectrum of antibacterial activity - some are ineffective against Gram-negatives while others have a very broad spectrum. Compared to my hypothetical review, this review seems superficial. On the other hand, it functions as a quick, broad survey of the field of resistance related to translation and protein synthesis.

At times I think it loses track of what it is supposed to be discussing in each section as well. The antifungal tRNA synthetase drug section in particular seems like it doesn't fit where it is. Likewise the long discussion of EF-P (pages 8-9) is at best of marginal interest to clinical antibiotics and resistance and could be greatly shortened or cut completely. Similarly for the long discussion of rpoD and mistranslation.

I get the impression that the authors have placed an artificial limitation on themselves to only discuss one or two examples of each phenomenon, which may be why the review often seems superficial. The section where this is a serious problem is in the New Frontiers in Antibiotic Development section where I don't think they can have this section with only odorhabdins as an example. The Pew Trusts database maintains a listing of all antibiotics in clinical trials development globally (current and past decade at least). I think the authors should consult this database and include in their review information on the multiple new aminoglycosides, tetracyclines, macrolides, oxazolidinones, etc that are being brought through clinical trials or have recently been approved for clinical use. For example, one interesting compound is a chemical hybrid of an oxazolidinone and quinolone, dual targeting translation and topoisomerase. This compound went all the way through Phase 3 clinical trials but development has recently been suspended by the company. Nevertheless, I think it is important to bring this hybrid up as an important example of a direction in which translation targeting antibiotics could go. It would also be important to discuss why new variants of classical classes are being developed - what problems are they addressing (in many cases it is connected with overcoming resistance mechanisms and so should be of interest for this review).

Here are some line comments:

Line 56: Does halting protein synthesis rapidly lead to elimination of bacteria? Chloramphenicol and Macrolides halt translation and are bacteriostatic.

Line 77 (Figure 1). Figure 1 serves to give a very general overview of important ribosomal locations for antibiotic action. I think however, that the authors should consider including either additional more detailed figures (or additional panels to this figure) showing the locations of interaction of several specific classes of antibiotics, including the odilorhabdins.

Line 116. It is not correct to state that all S12 mutations come with a significant fitness cost. Multiple publications have shown in various species, and in animal infections, that at least one S12 mutation is essentially cost-free. As an example I suggest you include reference to a paper by Erik Böttgers group showing a strong bias in favour of this mutation in resistant *Mycobacterium tuberculosis* clinical isolates: Böttger et al, (1998) *Nature Medicine* 4, 1343-1344.

Line 129: As I read them, both of the citations propose or support models that TetO and TetM dislodge bound tetracycline and induce a conformational change that prevents tetracycline rebinding after the RPP has left. I did not see anything that indicated evidence for direct competition for ribosomal binding between RPPs and tetracycline. I think you should rewrite this section to better reflect the experimental evidence.

Line 135: ABCF needs to be defined and introduced for the uninitiated. It seems strange to introduce them with their most important characteristic being that they are NTP hydrolases.

Line 159 - 160. I think you should give credit to the references with precedent for fusidic acid resistance mutations in *S. aureus*. I suggest you include reference to Nagaev et al, (2001) *Mol Microbiol*, 40, 433-439; and Laurberg et al (2000) *J Mol Biol* 303, 593-603.

Lines 186-219. I agree that EF-P is an interesting and important translation factor. However, it is not a target for antibiotics and I think this section could be entirely removed (or reduced to a very short comment on its potential as a future target).

Lines 221-248. Similarly to the above, mistranslation is interesting for ribosomologists but it is not a significant player in antibiotic therapy and this section should either be removed or reduced to a very short commentary on its potential for use.

Lines 236-239: In addition: This is very unclear to me and needs to be reworded to make the intended meaning clear. I think that some of the unclearness may come from the introduction of an 'antibiotic resistance protein' in the middle of this section.

Lines 272-280: This section about drugs targeting tRNA synthetases is out of place in a discussion of resistance mechanisms. It should be moved to the New Frontiers in Antibiotic Development section. I think that tRNA synthetases as novel drug targets should be discussed but you must be more comprehensive and include the examples of GSK2251052 and the recent GSK Tuberculosis tRNA synthetase inhibitors. You should also mention the one that has been on the market for decades - mupirocin.

The New Frontiers in Antibiotic Development section should be expanded. There have been other developments in translation targeting drugs other than odilorhabdins. Please examine the literature and especially the Pew Trust database (there are several aminoglycosides and tetracyclines and macrolides designed to overcome resistance mechanisms, for example).

Lines 331 - 346. It seems wrong to have this long speculation about modification of chloramphenicol while ignoring the actual aminoglycosides, tetracyclines, macrolides etc that are in clinical trials or recently approved.

Lines 347 - 355. Similar criticism to above - this text on modifications of neomycin is speculative ribosomology rather than a discussion of developments in antibiotic development. In a long review there would be a place for such speculations but not at the expense of ignoring all of the developments that are actually occurring.

Lines 356 - 365. Same comments as above - you ignore the actual macrolides and ketolides in development and instead return to discussing a potential resistance mechanism.

Lines 366 - 394. This section is very speculative and it would be better to devote text to actual examples where an adjuvant has been shown to increase the potency of an antibiotic. There are publications where adjuvants have for example been used to increase the entry of Gram-positive antibiotic (macrolides) into Gram-negatives. These have not made it to the clinical yet but I think it would be appropriate to discuss the potential of actual examples like this rather than purely speculative examples (like mistranslation/kasugomycin and EttA ADP/ATP ratios).

Lines 409 - 416. I don't think the discussion of persisters is relevant to the review and certainly is not a good way to end the review. Better to end with a summary of the major developments in drug discovery in the area and the problems they are trying to address.

Referee: 2

Comments to the Author(s)

This review by Witzky et al. provides an overview of current knowledge of bacterial resistance mechanisms against antibiotics that target the ribosome (and other components of the translational machinery), along with discussion of work towards or new potential strategies that might alleviate this important clinical problem. This significance of this topic is clearly high - antibiotics that target protein synthesis have been critical in treatment of bacterial infection but many (if not all) are threatened by one or more forms of resistance. The global threat of antibiotic resistance is widely acknowledged, making this review of interest to a potentially wide audience. As acknowledged by the authors, given the importance of these drugs and the topic in general, several other detailed reviews have been published in recent years that cover topics of translation-targeting antibiotics and their interactions with the ribosome, mechanism of action and resistance mechanisms. However, the approach taken by the authors here to overview resistance mechanisms together with substantial discussion of future perspectives on the opportunities and challenges faced in overcoming this resistance gives the work a unique perspective and makes it potentially a valuable contribution to this field.

While I am overall positive about the potential contribution of this work, to maximize its impact, I encourage the authors carefully review a number of aspects of how the material is presented. Three main aspects could be strengthened:

Eliminating a number of instances of vague or overly informal language, oversimplification of important concepts, and the odd example of hyperbole (do bacteria really "run rampant"?). While it is important for a review of this nature to be broadly accessible this should not come at the expense of an appropriate level of accuracy and rigor (some examples are included in the specific comments below).

A single, not especially useful figure is used to illustrate the entire discussion. This figure should feature earlier and could be substantially improved e.g. label binding sites of the antibiotics and

resistance mechanisms discussed (e.g. where modifications or mutation lead to resistance). All elements of the figure should be labeled (not just the subunits-mRNA, tRNA, nascent peptide). A second panel summarizing the discussion of aaRS targeting antibiotics and resistance mechanism would also be useful. More generally, many other aspects of the discussion would benefit from illustrations – these could be structures of proteins (such as ribosome bound protection factors) involved highlighting key features mentioned in the text, model figures summarizing mechanism of action, etc.

In the section on adjuvant strategies (or in its own sub-section) the approach of directly inhibiting resistance determinants (“resistance breakers”) should be included.

Specific comments and suggestions:

1. Line 56, “protein synthesis rapidly halts, leading to elimination of the bug”... how does the drug eliminate? Bacteriostatic vs bactericidal activity, etc. (I also generally object to the use of the term “bug”!)
2. Line 63, “bacteria rapidly evolving mechanisms of resistance”... not clear this entirely true; resistance mechanisms can be acquired by gene transfer and also (as later noted) there is compelling evidence for the ancient nature of resistance (pre-human use of antibiotics).
3. Line 75, specify “bacterial ribosome” (others are not 30S/50S)
4. Line 77,78 – include inducing miscoding as mechanism of action of aminoglycosides (this is mentioned later but should be stated here)
5. Line 82, “Bacteria will often employ inducible methyltransferases”... this is not so clear cut and could be described in a little more detail (how are they inducible, under what circumstances; drug producer vs. acquired resistance enzyme in a pathogen, etc)
6. Line 86-87, statement that ribosome modification “often” creates a significant impediment should be supported by references. Some elaboration may also be needed, e.g. it is not so clear cut that this is the case for all aminoglycoside resistance.
7. Line 133, grammatical issue? (suggest “preventing tetracycline binding and removing drug once bound”)
8. Line 156, “directly OR indirectly”
9. Line 165, mutations in rRNA, protein, both?
10. Line 168, this statement is misleading – EF-Tu hydrolyses GTP regardless of whether correct codon-anticodon pairing occurs.
11. Line 181, ribosomes are not “permanently” inactivated – competitive inhibitor with a dominant effect due to increased affinity.
12. Lines 212-220 (discussion of EF-P). Not really clear what the main take-away point is. here. What is the evidence for high translational demand? Does need to evade immune system or outcompete commensals depend on location/ type of infection/ pathogen? What about dormancy (also an effective strategy to evade antibiotic treatment, also immune system)? Do common commensals have EF-P and should this also be a consideration here?

13. Lines 229-239, I also found this description of RpsD mediated effects confusing. In line 236 why “also” maintains? (Isn’t the discussion about RpsD leading to mistranslation?) And, which “antibiotic resistance protein” is referred to in Line 237?

14. Line 252-253, “non-cognate” is by definition in “the incorrect location”.

15. Line 323, while likely true at least in part, interspecies competition is probably not the only reason for bacterial evolution of antibiotics (intra/ interspecies “communication”, regulation of gene expression, etc). The wording here is also a little confusing in terms of natural vs. semi-synthetic antibiotics: Line 324 appears refer to alteration of existing (natural) drugs; additionally, there is no “this method” described in the preceding sentence and also the next sentence refers to them as being “isolated” (which suggests the natural compounds themselves). Finally, the paragraph ends with the statement that efforts have turned to chemical modification (which is where it seemed start...).

16. Line 332, first reference to the figure (despite multiple prior mentions of tRNA binding sites or other features).

17. Line 338, instability of what? The drug itself? (Chemical lability, greater sensitivity to degradation). Is it shown that the alternative site isn’t just less effective for ribosome inhibition?

18. Line 339, “Derivatization of chloramphenicol can also... impact drug activity.” Isn’t this also the case for the previous example? If so, why “also”? (It’s not completely clear but I assume here “activity” refers to target or mechanism of action? However, even that changed in the previous example as the ribosome binding site was altered).

19. Phrase “Translation is an essential process” is repeated in Line 396 & 403.

Author's Response to Decision Letter for (RSOB-19-0051.R0)

See Appendix A.

RSOB-19-0051.R1 (Revision)

Review form: Reviewer 1

Recommendation

Accept as is

Are each of the following suitable for general readers?

- a) **Title**
Yes

b) **Summary**
Yes

c) **Introduction**
Yes

Is the length of the paper justified?

Yes

Should the paper be seen by a specialist statistical reviewer?

No

Is it clear how to make all supporting data available?

Not Applicable

Is the supplementary material necessary; and if so is it adequate and clear?

Not Applicable

Do you have any ethical concerns with this paper?

No

Comments to the Author

Dear Authors,

I am satisfied with the revisions you have made. I think the review reads very well and will be interesting to a wide audience.

Review form: Reviewer 2

Recommendation

Accept with minor revision (please list in comments)

Are each of the following suitable for general readers?

a) **Title**
Yes

b) **Summary**
Yes

c) **Introduction**
Yes

Is the length of the paper justified?

Yes

Should the paper be seen by a specialist statistical reviewer?

No

Is it clear how to make all supporting data available?

Not Applicable

Is the supplementary material necessary; and if so is it adequate and clear?

Not Applicable

Do you have any ethical concerns with this paper?

No

Comments to the Author

The authors have been responsive to the previous reviews and the the revised manuscript is much improved and strengthened by the additional descriptions and detail, particularly in the later sections. I have only a few very minor comments/ typos:

1. line 158, elucidation of the structures of
2. line 394, look for
3. line 400, and in murine and *Galleria mellonella* infection models
4. line 416, S-adenosyl-L-methionine
5. Figure 2 legend: title could be more descriptive of the point of the figure (e.g. "Ribosomal protection proteins bind the ribosome to dislodge inhibitory drugs and restore ribosome activity." Also, description for panel B - what "drug"?)
6. Figure 3 legend: each sentence could be improved grammatically; also what "dies"? (sentence subject appears to be the resistance protein but presumably mean bacteria expressing the protein)

Decision letter (RSOB-19-0051.R1)

07-Jun-2019

Dear Professor Ibba

We are pleased to inform you that your manuscript RSOB-19-0051.R1 entitled "Translational control of antibiotic resistance" has been accepted by the Editor for publication in Open Biology. The reviewer(s) have recommended publication, but also suggest some minor revisions to your manuscript. Therefore, we invite you to respond to the reviewer(s)' comments and revise your manuscript.

Please submit the revised version of your manuscript within 7 days. If you do not think you will be able to meet this date please let us know immediately and we can extend this deadline for you.

- 1) A text file of the manuscript (doc, txt, rtf or tex), including the references, tables (including captions) and figure captions. Please remove any tracked changes from the text before submission. PDF files are not an accepted format for the "Main Document".
- 2) A separate electronic file of each figure (tiff, EPS or print-quality PDF preferred). The format should be produced directly from original creation package, or original software format. Please note that PowerPoint files are not accepted.
- 3) Electronic supplementary material: this should be contained in a separate file from the main text and meet our ESM criteria (see <http://royalsocietypublishing.org/instructions-authors#question5>). All supplementary materials accompanying an accepted article will be treated as in their final form. They will be published alongside the paper on the journal website and posted on the online figshare repository. Files on figshare will be made available approximately one week before the accompanying article so that the supplementary material can be attributed a unique DOI.

Online supplementary material will also carry the title and description provided during submission, so please ensure these are accurate and informative. Note that the Royal Society will not edit or typeset supplementary material and it will be hosted as provided. Please ensure that the supplementary material includes the paper details (authors, title, journal name, article DOI). Your article DOI will be 10.1098/rsob.2016[*last 4 digits of e.g. 10.1098/rsob.20160049*].

- 4) A media summary: a short non-technical summary (up to 100 words) of the key findings/importance of your manuscript. Please try to write in simple English, avoid jargon, explain the importance of the topic, outline the main implications and describe why this topic is newsworthy.

Images

Data-Sharing

It is a condition of publication that data supporting your paper are made available. Data should be made available either in the electronic supplementary material or through an appropriate repository. Details of how to access data should be included in your paper. Please see <http://royalsocietypublishing.org/site/authors/policy.xhtml#question6> for more details.

Data accessibility section

Sincerely,
The Open Biology Team
mailto:openbiology@royalsociety.org

Reviewer(s)' Comments to Author:

Referee: 1

Comments to the Author(s)

Dear Authors,

I am satisfied with the revisions you have made. I think the review reads very well and will be interesting to a wide audience.

Referee: 2

Comments to the Author(s)

The authors have been responsive to the previous reviews and the the revised manuscript is much improved and strengthened by the additional descriptions and detail, particularly in the later sections. I have only a few very minor comments/ typos:

1. line 158, elucidation of the structures of
2. line 394, look for
3. line 400, and in murine and *Galleria mellonella* infection models
4. line 416, S-adenosyl-L-methionine
5. Figure 2 legend: title could be more descriptive of the point of the figure (e.g. "Ribosomal protection proteins bind the ribosome to dislodge inhibitory drugs and restore ribosome activity." Also, description for panel B - what "drug"?)
6. Figure 3 legend: each sentence could be improved grammatically; also what "dies"? (sentence subject appears to be the resistance protein but presumably mean bacteria expressing the protein)

Author's Response to Decision Letter for (RSOB-19-0051.R1)

See Appendix B.

Decision letter (RSOB-19-0051.R2)

11-Jun-2019

Dear Dr Ibba

We are pleased to inform you that your manuscript entitled "Translational control of antibiotic resistance" has been accepted by the Editor for publication in Open Biology.

Sincerely,

The Open Biology Team
mailto: openbiology@royalsociety.org

Appendix A

Department of Microbiology

484 West 12th Avenue
Columbus, OH 43210-1292

Phone (614) 292-2120
Fax (614) 292-8120
E-mail ibba.1@osu.edu

May 17, 2019

David M. Glover PhD ScD FRS FRSE
Director of Research and Wellcome Investigator
University of Cambridge
Department of Genetics
Cambridge CB2 3EH, UK

Dear David:

Thank you for your e-mail of April 1st concerning our manuscript "*Translational Control of Antibiotic Resistance*" by Witzky *et al.* which we submitted for publication in the *Royal Society's Open Biology* journal. We read your and the reviewers' comments with interest and fully agree that the manuscript would greatly benefit from the revisions suggested. Consequently, a number of changes have been made which are detailed below, with reference to the specific comments of yourself and each of the referees. Thank you for your help with this, and we hope that you now find our revised manuscript suitable for publication. We look forward to hearing from you.

Sincerely,

Michael Ibba, Ph.D.
Professor and Chair
Department of Microbiology

Referee: 1

Line 56: Does halting protein synthesis rapidly lead to elimination of bacteria? Chloramphenicol and Macrolides halt translation and are bacteriostatic.

This sentence has been reworded so it is clearer.

Line 77 (Figure 1). Figure 1 serves to give a very general overview of important ribosomal locations for antibiotic action. I think however, that the authors should consider including either additional more detailed figures (or additional panels to this figure) showing the locations of interaction of several specific classes of antibiotics, including the odilorhabdins.

We have modified this figure and added additional figures.

Line 116. It is not correct to state that all S12 mutations come with a significant fitness cost. Multiple publications have shown in various species, and in animal infections, that at least one S12 mutations is essentially cost-free. As an example I suggest you include reference to a paper by Erik Böttgers group showing a strong bias in favour of this mutation in resistant *Mycobacterium tuberculosis* clinical isolates: Böttger et al, (1998) *Nature Medicine* 4, 1343-1344.

We have added this information to the review.

Line 129: As I read them, both of the citations propose or support models that TetO and TetM dislodge bound tetracycline and induce a conformational change that prevents tetracycline rebinding after the RPP has left. I did not see anything that indicated evidence for direct competition for ribosomal binding between RPPs and tetracycline. I think you should rewrite this section to better reflect the experimental evidence.

This section has been clarified.

Line 135: ABCF needs to be defined and introduced for the uninitiated. It seems strange to introduce them with their most important characteristic being that they are NTP hydrolases.

This section has been clarified.

Line 159 - 160. I think you should give credit to the references with precedent for fusidic acid resistance mutations in *S. aureus*. I suggest you include reference to Nagaev et al, (2001) *Mol Microbiol*, 40, 433-439; and Laurberg et al (2000) *J Mol Biol* 303, 593-603.

These citations have been added.

Lines 186-219. I agree that EF-P is an interesting and important translation factor. However, it is not a target for antibiotics and I think this section could be entirely removed (or reduced to a very short comment on its potential as a future target).

This section has been moved to the adjuvant section and shortened.

Lines 221-248. Similarly to the above, mistranslation is interesting for ribosomologists but it is not a significant player in antibiotic therapy and this section should either be removed or reduced to a very short commentary on its potential for use.

Clinical isolates have been identified that have GatCAB and aaRS mutations. We have edited this section to emphasize the relevance of this form of resistance by directly mentioning these.

Lines 236-239: In addition: This is very unclear to me and needs to be reworded to make the

intended meaning clear. I think that some of the unclearness may come from the introduction of an 'antibiotic resistance protein' in the middle of this section.

We have modified this section to explicitly name the antibiotic resistance protein (TEM-1) earlier in the paragraph and clarify the intended meaning.

Lines 272-280: This section about drugs targeting tRNA synthetases is out of place in a discussion of resistance mechanisms. It should be moved to the New Frontiers in Antibiotic Development section. I think that tRNA synthetases as novel drug targets should be discussed but you must be more comprehensive and include the examples of GSK2251052 and the recent GSK Tuberculosis tRNA synthetase inhibitors. You should also mention the one that has been on the market for decades - mupirocin.

We have removed this section.

The New Frontiers in Antibiotic Development section should be expanded. There have been other developments in translation targeting drugs other than odilorhaddins. Please examine the literature and especially the Pew Trust database (there are several aminoglycosides and tetracyclines and macrolides designed to overcome resistance mechanisms, for example).

We have added the suggested information.

Lines 331 - 346. It seems wrong to have this long speculation about modification of chloramphenicol while ignoring the actual aminoglycosides, tetracyclines, macrolides etc that are in clinical trials or recently approved.

We have added the suggested information.

Lines 347 - 355. Similar criticism to above - this text on modifications of neomycin is speculative ribosomology rather than a discussion of developments in antibiotic development. In a long review there would be a place for such speculations but not at the expense of ignoring all of the developments that are actually occurring.

We have added the suggested information.

Lines 356 - 365. Same comments as above - you ignore the actual macrolides and ketolides in development and instead return to discussing a potential resistance mechanism.

We have added the suggested information.

Lines 366 - 394. This section is very speculative and it would be better to devote text to actual examples where an adjuvant has been shown to increase the potency of an antibiotic. There are publications where adjuvants have for example been used to increase the entry of Gram-positive antibiotic (macrolides) into Gram-negatives. These have not made it to the clinical yet but I think it would be appropriate to discuss the potential of actual examples like this rather than purely speculative examples (like mistranslation/kasugomycin and EttA ADP/ATP ratios).

We have added additional examples of adjuvants.

Lines 409 - 416. I don't think the discussion of persisters is relevant to the review and certainly is not a good way to end the review. Better to end with a summary of the major developments in drug discovery in the area and the problems they are trying to address.

We have shortened this part to de-emphasize it.

1. Line 56, “protein synthesis rapidly halts, leading to elimination of the bug”... how does the drug eliminate? Bacteriostatic vs bactericidal activity, etc. (I also generally object to the use of the term “bug”!)
“Elimination” was referring to the clearance of the pathogen from a host, not necessarily killing. This sentence has been reworded so it is clearer.
2. Line 63, “bacteria rapidly evolving mechanisms of resistance”... not clear this entirely true; resistance mechanisms can be acquired by gene transfer and also (as later noted) there is compelling evidence for the ancient nature of resistance (pre-human use of antibiotics).
This sentence has been modified to include “acquiring and evolving” mechanisms of resistance.
3. Line 75, specify “bacterial ribosome” (others are not 30S/50S)
We have made the suggested change in the text.
4. Line 77,78 – include inducing miscoding as mechanism of action of aminoglycosides (this is mentioned later but should be stated here)
We have made the suggested change in the text.
5. Line 82, “Bacteria will often employ inducible methyltransferases”... this is not so clear cut and could be described in a little more detail (how are they inducible, under what circumstances; drug producer vs. acquired resistance enzyme in a pathogen, etc)
We have added a more detailed description of the induction further into the paragraph.
6. Line 86-87, statement that ribosome modification “often” creates a significant impediment should be supported by references. Some elaboration may also be needed, e.g. it is not so clear cut that this is the case for all aminoglycoside resistance.
We have made the suggested change in the text.
7. Line 133, grammatical issue? (suggest “preventing tetracycline binding and removing drug once bound”)
We have made the suggested change in the text.
8. Line 156, “directly OR indirectly”
We have made the suggested change in the text.
9. Line 165, mutations in rRNA, protein, both?
We have specified ribosomal protein S12.
10. Line 168, this statement is misleading – EF-Tu hydrolyses GTP regardless of whether correct codon-anticodon pairing occurs.
We have removed the word “correct.”
11. Line 181, ribosomes are not “permanently” inactivated – competitive inhibitor with a dominant effect due to increased affinity.
We have made the suggested change in the text.
12. Lines 212-220 (discussion of EF-P). Not really clear what the main take-away point is here. What is the evidence for high translational demand? Does need to evade immune system or outcompete commensals depend on location/ type of infection/ pathogen? What about

dormancy (also an effective strategy to evade antibiotic treatment, also immune system)? Do common commensals have EF-P and should this also be a consideration here?

Li et al. 2019 established that salmonella must quickly undergo proteomic reprogramming to survive in a macrophage. We have modified this section to be more clear for the reader more precisely reflect this data. We have also addressed the dormancy concern.

13. Lines 229-239, I also found this description of RpsD mediated effects confusing. In line 236 why “also” maintains? (Isn’t the discussion about RpsD leading to mistranslation?) And, which “antibiotic resistance protein” is referred to in Line 237?

We have clarified this section by explicitly naming the antibiotic resistance protein, TEM-1. We have also removed the “also” on line 236.

14. Line 252-253, “non-cognate” is by definition in “the incorrect location”.

We have removed the phrase “the incorrect location”

15. Line 323, while likely true at least in part, interspecies competition is probably not the only reason for bacterial evolution of antibiotics (intra/ interspecies “communication”, regulation of gene expression, etc). The wording here is also a little confusing in terms of natural vs. semi-synthetic antibiotics: Line 324 appears refer to alteration of existing (natural) drugs; additionally, there is no “this method” described in the preceding sentence and also the next sentence refers to them as being “isolated” (which suggests the natural compounds themselves). Finally, the paragraph ends with the statement that efforts have turned to chemical modification (which is where it seemed start...).

We have clarified this section.

16. Line 332, first reference to the figure (despite multiple prior mentions of tRNA binding sites or other features).

We have added more references to the figure 1 and added additional figures.

17. Line 338, instability of what? The drug itself? (Chemical lability, greater sensitivity to degradation). Is it shown that the alternative site isn’t just less effective for ribosome inhibition?

This was added by mistake and has been removed.

18. Line 339, “Derivatization of chloramphenicol can also... impact drug activity.” Isn’t this also the case for the previous example? If so, why “also”? (It’s not completely clear but I assume here “activity” refers to target or mechanism of action? However, even that changed in the previous example as the ribosome binding site was altered).

We have clarified this sentence.

19. Phrase “Translation is an essential process” is repeated in Line 396 & 403.

We removed the first use of this phrase.

Appendix B

Response to Reviewers

Reviewer(s)' Comments to Author:

Referee: 1

Comments to the Author(s)

Dear Authors,

I am satisfied with the revisions you have made. I think the review reads very well and will be interesting to a wide audience.

Referee: 2

Comments to the Author(s)

The authors have been responsive to the previous reviews and the revised manuscript is much improved and strengthened by the additional descriptions and detail, particularly in the later sections. I have only a few very minor comments/ typos:

1. line 158, elucidation of the structures of

We have made the suggested change in the text.

2. line 394, look for

We have made the suggested change in the text.

3. line 400, and in murine and *Galleria mellonella* infection models

We have made the suggested change in the text.

4. line 416, S-adenosyl-L-methionine

We have made the suggested change in the text.

5. Figure 2 legend: title could be more descriptive of the point of the figure (e.g. "Ribosomal protection proteins bind the ribosome to dislodge inhibitory drugs and restore ribosome activity." Also, description for panel B - what "drug"?)

We have made the suggested change in the figure legend.

6. Figure 3 legend: each sentence could be improved grammatically; also what "dies"? (sentence subject appears to be the resistance protein but presumably mean bacteria expressing the protein)

We have made the suggested change in the figure legend.